# Association between occupational exposures and chronic low back pain: Protocol for a systematic review and meta-analysis

**Alexander Jahn**[1]\*, **Johan Hviid Andersen**[2,3], **David Høyrup Christiansen**[2,4,5],
**Andreas Seidler**[6], **Annett Dalbøge**[1,2]

**1** Danish Ramazzini Centre, Department of Occupational Medicine, Aarhus University Hospital, Aarhus, Denmark, **2** Department of Clinical Medicine, Aarhus University, Aarhus, Denmark, **3** Department of Occupational Medicine—University Research Clinic, Danish Ramazzini Centre, Goedstrup Hospital, Herning, Denmark, **4** Elective Surgery Centre, Silkeborg Regional Hospital, Silkeborg, Denmark, **5** Research, Regional Hospital Central Jutland, Viborg, Denmark, **6** Institute and Policlinic of Occupational and Social Medicine (IPAS), Faculty of Medicine, Technische Universität Dresden, Dresden, Germany

\* alexjn@rm.dk

**Data Availability Statement:** No datasets were generated or analysed during the current study. All relevant data from this study will be made available upon study completion.

## Abstract

### Introduction

The association between occupational mechanical exposures and chronic low back pain (LBP) has been widely studied, however, few systematic reviews have evaluated the evidence of an association. Furthermore, little is known of the impact of occupational psychosocial exposures on chronic LBP. The aim of this systematic review and meta-analysis is to study the association between occupational mechanical and psychosocial exposures and chronic LBP.

### Methods

The study will be conducted as a systematic review using another systematic review published in 2014 as basis and has been registered in the International Prospective Register of Systematic Reviews (PROSPERO) with registration number CRD42021281996. A systematic literature search will be performed in 6 scientific databases to identified potential relevant studies published after 2014. Studies will systematically be excluded through a screening process performed independently by 2 reviewers. Exposures will include occupational mechanical and psychosocial exposures, and outcome will include chronic LBP (LBP ≥3 months, "degenerative" diseases, and lumbosacral radiculopathy). Study population will include persons in or above working age, and study designs will comprise cohort and case-control studies. The quality of each included study will be methodologically assessed by 2 independent reviewers and level of evidence of an association will be graded using the Grading of Recommendations, Assessment, Development and Evaluations (GRADE) system. In meta-analyses, effect sizes will be addressed using random-effect models, sensitivity analyses will explore the robustness of the meta-analysis, and heterogeneity assessed.

**Funding:** This protocol for a systematic review and meta-analysis is funded by The Working Environment Research Fund, part of The Ministry of Employment, administered by the Danish Working Environment Authority. Grant no. 44-2020-09 20205100711. The funding body played no role in developing the protocol. URL of funder website: www.https://amff.dk/.

**Competing interests:** The authors have declared that no competing interests exist.

## Discussion

This systematic review and meta-analysis will assess the evidence available of the association between occupational mechanical and psychosocial exposures and chronic LBP. The review can provide essential knowledge on the association, exposure-response relationships, thresholds, which may pave the way for political decisions on the occupational environment and the labour market insurance policy.

## Introduction

Low back pain (LBP) is a frequent health problem in the general population [1]. LBP is defined as pain and discomfort located to the lumbar region and/or gluteal region; anatomically outlined from the 12th thoracic vertebra to the gluteal sulcus with or without radiating pain [2]. In the majority of cases, it is difficult to determine the specific structural cause of the pain [3]. In 2017, the global point prevalence of activity-limiting LBP was estimated to be 7.5% of the general population, indicating that approximately 577 million people are affected [4]. Although LBP often are short-term, 10–20% develop chronic LBP ($\geq$3 months) in a life course, gradually increasing with age [5, 6]. Furthermore, chronic LBP is one of the most common causes of years lived with disability [3, 7–9], a common reason for loss of work days, and the leading cause of early retirement from the labour market [7, 10, 11].

Risk factors for LBP comprise both occupational and non-occupational exposures [3]. In 2014, a systematic review was conducted by the Swedish council on Health Technology Assessment (SBU) on the association between occupational exposures and back disorders [12]. The authors found moderate evidence of an association between LBP and manual materials handling, lifting loads, non-neutral working posture, forward-bended posture, and whole body vibrations. In the updated review, non-neutral working posture, e.g., working with flexion and/or rotation and/or lateral flexion of the spine in a combination with lifting loads, were changed from limited to moderate evidence.

For lifting loads, a meta-analysis based solely on cohort studies showed an increased risk of 10–15% for LBP of exposed workers, when compared to non-exposed workers [13]. Another meta-analysis showed a significantly strong association between the combination of lifting loads and bending of the trunk and lumbosacral radiculopathy. The authors also found exposure-response relation between number of years bending the trunk 20° and lifting loads with lumbosacral radiculopathy as well as an association between heavy physically demanding work and lumbosacral radiculopathy [14].

Occupational psychosocial exposures also seem to be associated with LBP. In a recent meta-analysis, using data from 18 studies, the results suggested an overlap between psychosocial workplace factors associated with LBP in general (i.e., workload, job control, and social support) and those associated with chronic LBP [15]. Moreover, in an occupational cohort in 615 nurses, only low job security remained independently associated with LBP [16].

In general, the area of research on the association between occupational exposures and LBP is somewhat comprehensive, and to our knowledge no systematic reviews has investigated the association between occupational mechanical exposures and the risk of developing chronic LBP defined as pain in $\geq$3 months. Therefore, the proposed systematic review is an update of the SBU [12] restricted to occupational mechanical and psychosocial exposures for chronic LBP only. The aim of this systematic review and meta-analysis is to study

the association between occupational mechanical and psychosocial exposures and chronic LBP.

## Methods

### Protocol and registration

The protocol for the systematic review has been developed using the PRISMA-P 2015 checklist (Preferred Reporting Items for Systematic Reviews and Meta-Analyses) [17]. The PRISMA-P 2015 checklist is presented in the S1 Appendix and the protocol has been registered in the International Prospective Register of Systematic Reviews (PROSPERO) with registration number CRD42021281996.

### Literature search and eligibility criteria

The literature search will be conducted in two steps. First, we will use the SBU to screen for eligible articles. It consists of systematic literature searches on articles published between 1980 and the 10th of January 2014 [12]. Second, in collaboration with a librarian, we will perform the exact same systematic literature search as in the SBU carried out in the following databases: National Library of Medicine (PubMed), Excerpta Medica Database (EMBASE), PsycInfo, Cumulative Index to Nursing and Allied Health Literature (CINAHL), Cochrane Library, and Web of Science. Our literature search will include articles published from January the 10th 2014. Search terms are presented in the S2 Appendix using blocks with Medical Subject Headings (MeSH), truncations and phrases, depending on each database, using Boolean operators. Reference lists of included full text articles will be hand searched to identify potential relevant studies not found in the databases.

Our PECOS (Population, Exposure, Comparison, Outcome, and Study design) are presented in S3 Appendix. The criteria for the study population will be restricted to adults in or above working age with no limitations regarding participants' sex, demographics, or ethnicity. Exposures will include occupational mechanical exposures (e.g., lifting or carrying of load, working postures, and vibrations) and psychosocial exposures (e.g., job strain, effort reward imbalance and organizational injustice). To ensure consistency in type of mechanical exposures, proxy estimates of exposures such as job titles, physical activity in leisure time, sports, and other athletic activities will be excluded. The outcome will include chronic LBP defined as pain for ≥3 months or an indicator of chronic LBP (i.e., "degenerative" diseases or lumbosacral radiculopathy). If no indication of time with LBP or information on an indicator of chronicity is provided, the study will be excluded. Studies investigating accidents/injuries, inherent pain or pain caused by other diseases not related to occupational mechanical exposures, and proxy measurements to chronic LBP (e.g., sickness absenteeism) will be excluded. Only studies investigating the association between occupational exposures and chronic LBP expressed in appropriate risk estimates will be included–or if appropriate risk estimates are possible to calculate. Study designs will comprise observational epidemiological studies where adequate evidence of causal associations are possible to interpret, therefore excluding cross-sectional designs. Finally, only peer-reviewed articles in English, Danish, Swedish, and Norwegian will be included due to language barriers; hence, this review will not consider conference abstracts or dissertations.

Using the review-management software Covidence and reference-management software EndNote 20, all duplicates will be removed before 2 reviewers independently will perform the exclusion of irrelevant studies in Covidence. The 2 reviewers will exclude studies in 2 steps, first screening on title/abstract followed by full-text screening. Discrepancies between the 2 reviewers will be solved through discussion until consensus is reached by all authors.

## Data extraction, assessment of methodological study quality, and evidence of an association

Two reviewers will conduct the data extraction from each included study entering the information in a pre-defined table. Overall, the data extraction will be divided into a descriptive section and a section including all effect sizes. The data extraction will include information about author, exposures, outcomes, demographic characteristics of included population (e.g., age, sex) and study characteristics (e.g., sample size, country, measurements such as effect size). With emphasise on the exposure, data extraction will include origin of measurement, exposure dimension (e.g., duration, intensity), and distinguishing between objective or subjective measures along with compositions of exposure groups. Same procedure will be applicable for outcome measures. Discrepancies between data chosen for the extraction will be solved through discussion in the author group until consensus is reached.

For the overall assessment of risk of bias, the authors agreed that study population and follow-up, exposure assessment, outcome assessment and the statistical analyses will have the most relevant impact on the assessment of risk of bias. To accommodate these impacts on the risk of bias, the methodological quality of the included studies will be critically appraised with a risk of bias tool adapted for the current scientific research question. The critical appraisal tool will be structured into fixed domains as described by Cochrane [18] and is presented in the S4 Appendix. The domains are based on previously established checklists used in research on chronic diseases used in several systematic reviews [14, 19–22]. The critical appraisal tool also considers criteria from the Critical Appraisal Skills Program (CASP) [23], Newcastle-Ottawa Scale [24], and Scottish Intercollegiate Guidelines Network (SIGN) [25] tools. The domains are divided into 5 major and 3 minor domains and the overall quality assessment of the included studies will comprise 3 categories: "low", "moderate", and "high" risk of bias. If a study is considered as low risk, all 5 major domains and at least 1 minor domains has to be rated as low risk. To be considered as moderate risk, 4 major domains and at least 1 minor domain has to be rated as low risk. All other combinations are considered as high risk of bias. The critical appraisal tool will be pilot-tested by the author group. Assessing the risk of bias of the included studies will be assessed by 2 independently review authors and their ratings will be compared. If disagreement occurs between ratings, this will be discussed in the review group until consensus is reached.

Publication bias will be investigated using funnel plots by Eggers test following recommendations from Sterne et al. [26] and the impact on the results discussed.

The overall strength of the evidence of an association will be evaluated in accordance with Grading of Recommendations, Assessment, Development and Evaluations (GRADE) [27]. Since GRADE was developed to provide methodological guidance for rating quality in reviewing intervention research, an adapted version of GRADE to prognostic factor research will be applied as proposed by Huguet et al. [28]. Two reviewers will independently rate the strength of evidence across studies where, initially, all studies are considered as high and can be downgraded based on their certainty of evidence. The domains for downgrading are phase of investigation, study limitations, inconsistency, indirectness, imprecision, and publication bias. Conversely, certainty in the evidence can be upgraded by increasing the confidence in the evidence, which is based upon the magnitude of effect, clear dose-response gradient or when residual confounding decreases the magnitude of effect.

## Statistics

Inter-rater agreement will be calculated to assess consistency between review authors quality rating based on the aforementioned 3 categories of risk of bias. To be included in the meta-

analyses, studies must provide sufficient quantitative information on effect sizes with 95% confidence intervals (95% CI) on the association between occupational exposures and chronic LBP. If this information is lacking or the reported effect sizes cannot be transformed, the relevant studies will be excluded. We will extract adjusted estimates from the individual studies, unless the estimates are adjusted for an exposure resulting in an over-adjustment as well as other relevant descriptive statistics. If a study population appears more than once among the included studies for the meta-analyses, exclusion will be based by the lowest quality and smallest sample size to avoid double-counting data.

The meta-analytic approach will be conducted by reporting Odds Ratios (OR) with a 95% CI through a synthesis of effect size measurements illustrated by forest plots. We expect substantial heterogeneity in, especially, exposure assessments across the included studies. Taking this into account, random-effects model will be used for the meta-analysis since different studies cannot be assumed to provide estimates of a common, true effect [29]. We will use Q and $I^2$ to identify and measure heterogeneity. Q will assess if heterogeneity in the effect sizes is statistically significant and $I^2$ (percentage of the variability in effect estimates that is due to heterogeneity) will quantify heterogeneity in the meta-analysis and be evaluated on the basis of Cochrane's guide to interpretation [30]. To further address heterogeneity, subgroup analyses will be conducted based on exposure assessments, and if necessary, exclusion of studies will be discussed in the author group if exposure assessment is so different that they should not be combined. Finally, if agreed upon in the author group, it could be possible to abstain from the meta-analysis and explore the heterogeneity instead.

If studies report effect sizes from independent subgroups within a study, subgroups will be treated as no different from independent studies (unique sample) to compute summary effect. If possible, effect sizes will be investigated for an exposure-response relationship for studies reporting on more than 2 occupational exposure categories. Furthermore, sensitivity analyses will be conducted restricting the meta-analysis to studies rated only as "high quality" exploring the robustness of the meta-analysis as well as a "leave-one-out" analysis determining the importance of individual studies.

## Discussion

This comprehensive systematic review and meta-analyses aim to update the existing review from SBU targeting specific occupational exposures and chronic LBP. By excluding cross-sectional studies and focusing on cohort and case-control designs, this review aims to evaluate potential causal relations between occupational exposures and development chronic LBP. Gathering and summarising information on psychosocial exposures can further enhance our understanding on occupational exposure.

Using an existing critical appraisal tool developed for chronic diseases and incorporating wordings from validated acknowledgeable tools, it is possible to minimise bias in ratings of included studies in contrast to using a generic tool. Furthermore, eliminating summarising scores used by other critical appraisal tools and introducing domains instead, we can address major risk of bias problems focussing on our specific research question. Evaluating the strength of epidemiological evidence, this review is the foundation of a scientific reference document that attempts to provide specific guidelines (thresholds) in regard to occupational mechanical exposures. We will consider a broad range of mechanical exposures, making it a major strength in providing insights into potential long term adverse effects of occupational exposures. Thereto, it is the first review specifically investigating the associations between occupational mechanical exposures and chronic LBP. Last but not least, the summarising information from this systematic review can provide essential knowledge for politicians in

regards of working environment and labour market insurance policy. Thereto, the information will be beneficial for employers, if addressing these potential adverse exposures, to increase health and well-being of the employees.

## Limitations

The effect of updating SBU's literature search has its limitations considering differences in inclusion and exclusion criteria. Firstly, even though the literature search is identical, the proposed systematic review will alter the criteria's for including studies. Thus, only a part of the SBU will be updated leaving out other exposures (e.g., air pollution, noise and chemicals) to be examined and updated at a later time. These differences in inclusion and exclusion criteria's is given in the additional file 3.

Secondly, the quality assessment will not be based on the same critical appraisal tool used in the SBU. We considered the SBU tool to be comprehensive, reducing the feasibility of the study. To account for this deviation, two reviewers will rate the studies included in the SBU using our developed methodological quality assessment tool. Thirdly, by not including cross-sectional design, it creates the possibility of not comprising all possible information, potential leaving out essential knowledge, but minimizing risk of reverse causality. Conversely, the meta-analysis aggregated effect sizes can provide an opportunity for causal relationships to be interpreted when including only longitudinal designs.

## Supporting information

**S1 Appendix. PRISMA-P 2015 checklist.**
(PDF)

**S2 Appendix. Search strategy.**
(PDF)

**S3 Appendix. PECOS.**
(PDF)

**S4 Appendix. Critical appraisal tool.**
(PDF)

## Author Contributions

**Funding acquisition:** Johan Hviid Andersen, David Høyrup Christiansen, Annett Dalbøge.

**Methodology:** Alexander Jahn.

**Project administration:** Alexander Jahn, David Høyrup Christiansen, Annett Dalbøge.

**Supervision:** Johan Hviid Andersen, David Høyrup Christiansen, Andreas Seidler, Annett Dalbøge.

**Writing – original draft:** Alexander Jahn.

**Writing – review & editing:** Johan Hviid Andersen, David Høyrup Christiansen, Andreas Seidler, Annett Dalbøge.

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
